# Parsimonious Quantile Regression of Financial Asset Tail Dynamics via Sequential Learning

**Xing Yan[3]**    **Weizhong Zhang[1]**    **Lin Ma[1]**    **Wei Liu[1]**    **Qi Wu[2,*]**

[1]Tencent AI Lab
[2]School of Data Science, City University of Hong Kong
[3]Department of SEEM, The Chinese University of Hong Kong

xyan@se.cuhk.edu.hk   {zhangweizhongzju,forest.linma}@gmail.com
wl2223@columbia.edu   qiwu55@cityu.edu.hk

## Abstract

We propose a parsimonious quantile regression framework to learn the dynamic tail behaviors of financial asset returns. Our model captures well both the time-varying characteristic and the asymmetrical heavy-tail property of financial time series. It combines the merits of a popular sequential neural network model, i.e., LSTM, with a novel parametric quantile function that we construct to represent the conditional distribution of asset returns. Our model also captures individually the serial dependences of higher moments, rather than just the volatility. Across a wide range of asset classes, the out-of-sample forecasts of conditional quantiles or VaR of our model outperform the GARCH family. Further, the proposed approach does not suffer from the issue of quantile crossing, nor does it expose to the ill-posedness comparing to the parametric probability density function approach.

## 1   Introduction

In general, machine learning models aim to predict one single value of output variable $y$ given input $x$, usually to estimate the conditional mean $E[y|x]$. In many situations, we are also interested in the characteristics of the conditional distribution $p(y|x)$. A typical domain needing the learning of these characteristics is financial returns. Data from financial markets is highly stochastic or noisy. It is impossible to accurately predict future financial returns. What we can predict and what we really care about are their conditional distributional characteristics like volatility, heavy tails, and Value-at-Risk, which are all widely used measures of risks. The huge and increasing demands for risk management and for understanding market behaviors make it extremely important to predict these characteristics.

In the scope of discrete-time econometric models, the benchmark of forecasting conditional distribution of time-$t$ asset return $r_t$ conditional on past return history is the Generalized Autoregressive Conditional Heteroskedasticity (GARCH) model as well as its variants. First appearing in [9] and [2] to model time-varying volatility, GARCH-type models have now become a big family, including popular variants like EGARCH [24], GJR-GARCH [16], TGARCH [32], etc. They all describe the distribution $p(r_t|r_{t-1}, r_{t-2}, \dots)$ by making strong assumptions on the probability density function of it, e.g., assuming it is Gaussian, and let the distribution parameters depend on past information. Usually, $t$-distribution is assumed to model heavy tails. Quantile regression [19][20] is another type of method to forecast the conditional distributional characteristics. It predicts the quantiles of $p(y|x)$ without making any distributional assumption. In this paper, we model and predict conditional quantiles and heavy tails of financial return series in a parsimonious quantile regression framework that describes the distribution $p(r_t|r_{t-1}, r_{t-2}, \dots)$ in a parametric quantile function way.

---

It is known that financial asset returns are heavy-tail distributed, both conditionally and unconditionally [7]. This has important consequences for both the pricing of assets and the management of their risks [15]. More importantly, their tail behaviors are not only asymmetrical but also time-varying. In GARCH family, $t$-distribution is heavy-tailed but symmetric, and more critically, the degrees of freedom which control the tail heaviness cannot vary with time. Previous studies [17][22][1][25] modelled time-varying conditional skewness and kurtosis in an autoregressive way, like volatility modelling in GARCH. However, they all assumed complicated probability density functions, some of which even have no analytical forms, which make model estimation difficult. Besides, models that allow the data to speak for itself rather than being restricted to linear auto-regressiveness are needed.

In this paper, we approach the problem by parameterizing the conditional quantile function of asset returns, instead of assuming they are drawn from a given probability density function, hence suffering from tractability and ill-posedness. Apart from probability density function, quantiles are another representation of the asymmetry and tailedness of a distribution. If one can model and estimate conditional quantiles for a fine set of probability levels, it achieves almost the same effect as modelling conditional mean, volatility, skewness, and kurtosis simultaneously. Quantile regression has the potential to undertake this interesting task, but the traditional version suffers from some issues. One is the lack of monotonicity in the estimated quantiles, also known as quantile crossing, despite some imperfect solutions proposed in [27] and [5]. Other issues include an increasing number of parameters when estimating more quantiles, and the lack of interpretability. Recently, some works deal with the large-scale [31] and high-dimensional [26] situations of quantile regression.

In this paper, we propose a parametric heavy-tailed quantile function (HTQF) to model a distribution with asymmetric left and right heavy tails. The Q-Q plot of the proposed HTQF against the standard normal distribution is of an inverted S shape, and the degrees of the tail heaviness are controlled by two parameters in a flexible way. Our HTQF overcomes the disadvantages of the probability density function approach in GARCH-type models when modelling asymmetric heavy tails. For financial asset returns, we let the quantile function of $p(r_t|r_{t-1}, r_{t-2}, \dots)$ be an HTQF and let the parameters of it be time-varying and depend on past information through a Long Short-term Memory (LSTM) unit [18], which is a popular sequential neural network model and has been applied to many practical problems like video understanding [11][29][30], video prediction [4], and video retrieval [12]. Parameters of the LSTM can be learned in a quantile regression framework with multiple probability levels. After training, the conditional quantiles of $r_t$ and the interpretable parameters of HTQF representing tail heaviness can be estimated.

Our model has significant advantages over GARCH-type models and traditional quantile regression. To summarize, our contributions are: (1) We propose a novel parametric quantile function to represent a distribution with asymmetric heavy tails, and leverage it to model the conditional distribution of financial return series. (2) In the quantile regression framework, coupled with an LSTM unit, our method can learn the time-varying tail behaviors successfully and predict conditional quantiles more accurately, as verified by our experiments. (3) We overcome the disadvantages of traditional quantile regression, including the quantile crossing, the increasing number of parameters when estimating more quantiles, and the lack of interpretability.

## 2 GARCH-type Models

For a univariate financial return series $\{r_t\}$, GARCH model was first proposed in [9] and [2] to model its time-varying volatility or volatility clustering. By making a prior assumption on the conditional distribution of the residual $\varepsilon_t = r_t - \mu_t$ ($\mu_t$ is the conditional mean of $r_t$), letting it be normal $\mathcal{N}(0, \sigma_t^2)$, the time-$t$ volatility $\sigma_t$ is modelled to depend on past residuals $\varepsilon_{t-1}, \dots, \varepsilon_{t-q}$ and past volatilities $\sigma_{t-1}, \dots, \sigma_{t-p}$. Formally, GARCH($p, q$) is specified as follows:

$$r_t = \mu_t + \varepsilon_t, \qquad \varepsilon_t|\psi_{t-1} \sim \mathcal{N}(0, \sigma_t^2), \tag{1}$$

$$\sigma_t^2 = \omega + \alpha_1\varepsilon_{t-1}^2 + \cdots + \alpha_q\varepsilon_{t-q}^2 + \beta_1\sigma_{t-1}^2 + \cdots + \beta_p\sigma_{t-p}^2, \tag{2}$$

where $\psi_{t-1}$ denotes the past information set. The parameters $\omega, \alpha_i, \beta_j$ can be estimated with maximum likelihood method. Because of its success in modelling and forecasting conditional volatility, a lot of extensions and variants had been proposed such as EGARCH [24], GJR-GARCH [16], TGARCH [32], etc. Most of them made reasonable and interpretable changes to Equation (2) and achieved better performances.

Alternatives can also be made in Equation (1). A better choice of the distribution assumption is Student's $t$-distribution: $\varepsilon_t = \sigma_t z_t$, $z_t|\psi_{t-1} \sim t(\nu)$, where $\nu$ is the degrees of freedom. $t$-distribution has symmetric heavy tails at the left and right sides. We denote GARCH-type models with the $t$-distribution assumption by GARCH-$t$, EGARCH-$t$, etc. Besides, one can also choose different ways to model the conditional mean $\mu_t$, e.g., to use GARCH-type models alone, it can be set to a constant $\mu_t = \mu$. One can also adopt the linear autoregressive way: $\mu_t = \gamma_0 + \gamma_1 r_{t-1} + \cdots + \gamma_s r_{t-s}$. We denote GARCH-type models with this linear autoregressive specification of conditional mean and with the $t$-distribution assumption by AR-GARCH-$t$, AR-EGARCH-$t$, etc.

Although GARCH-type models were initially designed to model and forecast conditional volatility, they can naturally be used to predict conditional quantiles because they fully describe the conditional distribution. Actually they are widely employed in finance to predict Value-at-Risk (VaR), which are the left-tail side quantiles, e.g., 0.01 or 0.05-quantile, representing downside risk of asset prices.

Another big family of models that have similarities with GARCH-type models are stochastic volatility (SV) models. Some comparisons between GARCH-type and SV models were made in [28][13][3][14]. SV models are applied in situations when volatility contains independent risk driver. In continuous time, if driven by Brownian Motion, they are Markovian, which is essentially different from GARCH family and our proposed model and may not be suitable for modelling serial dependence of volatility. What are comparable with GARCH-type and our models and are consistent with the focus of this paper, are long-memory volatility models driven by, e.g., fractional Brownian Motion or Hawkes process, and preferably in discrete time. CAViaR [10] is another similar model for estimating conditional quantiles inspired by GARCH. It models conditional quantiles separately for different probability levels instead of making assumptions on the full conditional distribution. So it is somewhat difficult to estimate the conditional moments, also different from GARCH-type and our models.

## 3 Traditional Quantile Regression

Quantiles are important characteristics of a distribution. For a continuous distribution density $p(y)$, for a given probability level $\tau \in (0, 1)$, e.g., $\tau = 0.1$ or $0.9$, the $\tau$-quantile $q$ of $p(y)$ is defined as $q = F^{-1}(\tau)$ where $F(y)$ is the cumulative distribution function of $p(y)$. Quantile regression [19][20] aims to estimate the $\tau$-quantile $q$ of the conditional distribution $p(y|x)$. To do this, without making any assumption on $p(y|x)$, a parametric function $q = f_\theta(x)$ is chosen, for example, a linear one $q = w^\top x + b$. Note that $q$ is an unobservable quantity, a specially designed loss function (named pinball loss in this paper) between $y$ and $q$ makes the estimation feasible in quantile regression:

$$L_\tau(y, q) = \begin{cases} \tau|y - q| & y > q \\ (1 - \tau)|y - q| & y \leq q \end{cases}. \tag{3}$$

Then we minimize the expected loss in a traditional regression way to get the estimated parameter $\hat{\theta}$:

$$\min_\theta \mathbf{E}[L_\tau(y, f_\theta(x))]. \tag{4}$$

Given a dataset $\{x_i, y_i\}_{i=1}^N$, the empirical average loss $\frac{1}{N} \sum_{i=1}^N L_\tau(y_i, f_\theta(x_i))$ is minimized instead. When we want to estimate multiple conditional quantiles $q_1, q_2, \ldots, q_K$ for different probability levels $\tau_1 < \tau_2 < \cdots < \tau_K$, $K$ different parametric functions $q_k = f_{\theta_k}(x)$ are chosen and the losses are summed up to be minimized simultaneously:

$$\min_{\theta_1, \ldots, \theta_K} \frac{1}{K} \frac{1}{N} \sum_{k=1}^K \sum_{i=1}^N L_{\tau_k}(y_i, f_{\theta_k}(x_i)). \tag{5}$$

However, this combination may lead to an embarrassing issue called quantile crossing, i.e., for some $x$ and $\tau_j < \tau_k$, it is possible that $f_{\theta_j}(x) > f_{\theta_k}(x)$ which contradicts the probability theory. It occurs because $\theta_j$ and $\theta_k$ are in fact independently estimated in the optimization. To overcome this, additional constraints on the monotonicity of the quantiles can be added to the optimization to ensure non-crossing [27]. Another simpler solution is post-processing, i.e., sorting or rearranging the original estimated quantiles to be monotone [5]. Another two shortcomings of this traditional quantile regression include an increasing number of parameters when estimating quantiles for a larger set of $\tau$, i.e., $K$ is larger. For a more elaborate description of a distribution, large $K$ is necessary in some cases. The other shortcoming is that the explicit mapping from $x$ to the conditional quantile has no interpretability, making it difficult to combine domain knowledge.

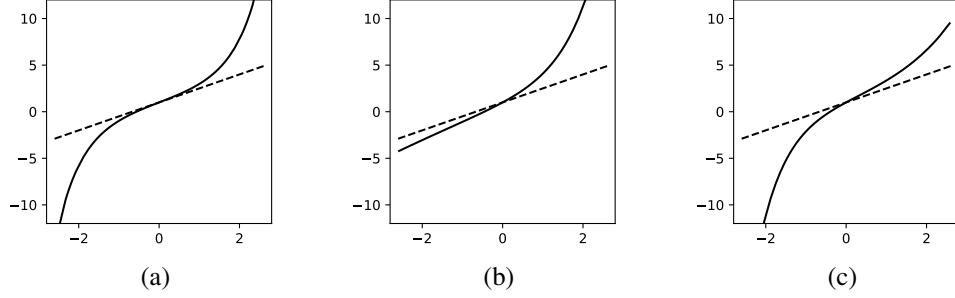

Figure 1: Q-Q plots against $\mathcal{N}(0,1)$: (a) $t(2)$; (b) HTQF with $u = 1.0$ and $v = 0.1$; (c) HTQF with $u = 0.6$ and $v = 1.2$. For all three distributions, $\mu = 1$ and $\sigma = 1.5$. For HTQF, $A = 4$.

## 4   Our Model

We first describe the proposed parametric quantile function, then show how it is used to model the conditional distribution $p(r_t|r_{t-1}, r_{t-2}, \dots)$ of financial return series and how the dependence on past information is modelled. Our proposed model is completed in a quantile regression framework.

### 4.1   Heavy-tailed Quantile Function

There are three common ways to fully express a continuous distribution, through probability density function (PDF), cumulative distribution function (CDF), or quantile function. In financial data modelling, much attention is paid to how to choose an appropriate parametric PDF that is consistent with the empirical facts of financial returns, like heavy tails. In our model, we design a parametric quantile function that allows varying tails and is intuitively easy to be understood.

Our idea starts from the Q-Q plot, which is a popular method to determine whether a set of observations follows a normal distribution or not. The theory behind this is quite simple: the $\tau$-quantile of a normal distribution $\mathcal{N}(\mu, \sigma^2)$ is $\mu + \sigma Z_\tau$, where $Z_\tau$ is the $\tau$-quantile of the standard normal one. When $\tau$ takes different values in $(0, 1)$, their Q-Q plot forms a straight line. If the Q-Q plot yields an inverted S shape, it indicates that the corresponding distribution is heavy-tailed (see Figure 1 (a) for an example of the Q-Q plot of $t$-distribution with 2 degrees of freedom against $\mathcal{N}(0, 1)$).

We construct a parsimonious parametric quantile function, as a function of $Z_\tau$, to let it have a controllable shape in the Q-Q plot against the standard normal distribution. Specifically, the up tail and down tail of the inverted S shape in the Q-Q plot are controlled by two parameters respectively. Our proposed heavy-tailed quantile function (HTQF) has the following form:

$$Q(\tau|\mu, \sigma, u, v) = \mu + \sigma Z_\tau \left( \frac{e^{uZ_\tau}}{A} + 1 \right) \left( \frac{e^{-vZ_\tau}}{A} + 1 \right), \qquad (6)$$

where $\mu$, $\sigma$ are location and scale parameters respectively, $A$ is a relatively large positive constant. $u > 0$ controls the up tail of the inverted S shape, i.e., the right tail of the corresponding distribution. $v > 0$ controls the down tail, i.e., the left tail of the corresponding distribution. The larger $u$ or $v$, the heavier the tail. When $u = v = 0$, the HTQF becomes the quantile function of a normal distribution.

To understand these, note that in Equation (6), $Z_\tau$ is first multiplied by two factors $f_u(Z_\tau) = e^{uZ_\tau}/A + 1$ and $f_v(Z_\tau) = e^{-vZ_\tau}/A + 1$, then multiplied by $\sigma$ and added by $\mu$ (for simplicity one can set $\mu = 0$ and $\sigma = 1$). The factor $f_u$ is a monotonically increasing and convex function of $Z_\tau$, and satisfies $f_u \to 1$ as $Z_\tau \to -\infty$. So $Z_\tau f_u(Z_\tau)$ will exhibit the up tail of the inverted S only. The same analysis applies to the factor $f_v$ too. Thus, $Z_\tau f_u(Z_\tau) f_v(Z_\tau)$ exhibits the whole inverted S of the Q-Q plot. The roles of $A$ are to let $f_u(0)$ and $f_v(0)$ be close to 1, and to ensure the HTQF is monotonically increasing with $Z_\tau$. Figure 1 (b) and (c) show the Q-Q plots of HTQF with different values of $u$ and $v$ against $\mathcal{N}(0, 1)$. They exhibit different degrees of tailedness and the tails can flexibly change according to $u$ and $v$. In addition, for an HTQF with fixed values of its parameters, there exists a unique probability distribution associated with it because the inverse function of it exists and is a CDF. Please refer to the proof in the supplementary material.

## 4.2 Quantile Regression with HTQF

For the distribution $p(r_t|r_{t-1}, r_{t-2}, \dots)$, different from GARCH-type models, we do not make assumptions on the PDF of it. Instead, we assume its quantile function being an HTQF, denoted by $Q(\tau|\mu_t, \sigma_t, u_t, v_t)$, where $\mu_t, \sigma_t, u_t, v_t$ are time-varying parameters representing the location, scale, and heavy tails of the corresponding distribution. Hence, the conditional $\tau_k$-quantile of $r_t$ can be easily obtained by putting $\tau_k$ into the function: $q_k^t = Q(\tau_k|\mu_t, \sigma_t, u_t, v_t), k = 1, \dots, K$.

Obviously, the parameters $\mu_t, \sigma_t, u_t, v_t$ should depend on past series $r_{t-1}, r_{t-2}, \dots$. To model that, we select a subsequence of fixed length from $r_{t-1}, r_{t-2}, \dots$ to construct a feature vector sequence, and apply an LSTM unit on it. LSTM [18] is a popular and powerful sequential neural network model in machine learning, so it is a natural choice in our method (see the supplementary material for a brief introduction and [23] for a comprehensive review of LSTM). In detail, a fixed length $L$ is chosen, and then a feature vector sequence of length $L$ is constructed from $r_{t-1}, \dots, r_{t-L}$:

$$x_1^t, \dots, x_L^t = \begin{bmatrix} r_{t-L} \\ (r_{t-L} - \bar{r}_t)^2 \\ (r_{t-L} - \bar{r}_t)^3 \\ (r_{t-L} - \bar{r}_t)^4 \end{bmatrix}, \dots, \begin{bmatrix} r_{t-1} \\ (r_{t-1} - \bar{r}_t)^2 \\ (r_{t-1} - \bar{r}_t)^3 \\ (r_{t-1} - \bar{r}_t)^4 \end{bmatrix}, \tag{7}$$

where $\bar{r}_t = \frac{1}{L}\sum_{i=1}^{L} r_{t-i}$. The intuition behind this construction is straightforward, which is to extract information contained in raw quantities associated with the first, second, third, and fourth central moments of past $L$ samples. After this construction, we model the four HTQF parameters $\mu_t, \sigma_t, u_t, v_t$ as the output of an LSTM unit when feeding input $x_1^t, \dots, x_L^t$:

$$[\mu_t, \sigma_t, u_t, v_t]^\top = \tanh(W^o h_t + b^o), \qquad h_t = \text{LSTM}_\Theta(x_1^t, \dots, x_L^t), \tag{8}$$

where $\Theta$ is the LSTM parameters, $h_t$ is the last hidden state. $W^o, b^o$ are the output layer parameters.

At last, for multiple probability levels $0 < \tau_1 < \tau_2 < \dots < \tau_K < 1$, the pinball losses between $r_t$ and its conditional quantiles $q_k^t = Q(\tau_k|\mu_t, \sigma_t, u_t, v_t)$ are summed up to be minimized together, like in traditional quantile regression:

$$\min_{\Theta, W^o, b^o} \frac{1}{K} \frac{1}{T-L} \sum_{k=1}^{K} \sum_{t=L+1}^{T} L_{\tau_k}\left(r_t, Q(\tau_k|\mu_t, \sigma_t, u_t, v_t)\right). \tag{9}$$

Combine Equation (6)(7)(8)(9) to complete our proposed quantile regression model using LSTM and HTQF, denoted by LSTM-HTQF. After training, for new subsequent series $\{r_{t'}\}$, the time-varying parameters $\mu_{t'}, \sigma_{t'}, u_{t'}, v_{t'}$ of HTQF can be calculated directly with the learned model parameters $\hat{\Theta}, \hat{W}^o, \hat{b}^o$. Among them, $\{u_{t'}\}$ and $\{v_{t'}\}$ can represent how the tails behave temporally. In addition, conditional quantiles $q_k^{t'}$ can be predicted and the summed loss in Equation (9) is evaluated again for testing the performance on the new subsequent series, since no ground truth of quantiles are available.

## 4.3 Discussions

The advantages of our model over GARCH-type models are obvious. The proposed HTQF is more intuitive and flexible to model asymmetric heavy tails than the PDFs in GARCH-type models, like the skewed generalized $t$-distribution in [1]. To have varying tails, one PDF must be in complicated analytical form that will make model estimation difficult. Even for the simplest one, the $t$-distribution, the analytical complexity of its PDF makes model estimation unfeasible if one assumes time-varying degrees of freedom, while our HTQF parameters can be easily set to be time-varying in the quantile regression framework. Besides, the LSTM can help to learn nonlinear dependence on past information while the linear auto-regressiveness in GARCH-type models cannot. We quantitatively compare our model to several classical GARCH-type models in the experiments.

Comparing to traditional quantile regression, our model overcomes the three shortcomings mentioned in Section 3. First, it is not hard to prove that the HTQF is a monotonically increasing function with $Z_\tau$, and also with $\tau$, so quantile crossing will never happen. Then, no matter how large $K$ is, i.e., a lot of quantiles need to be estimated, we only need HTQF's four parameters $\mu_t, \sigma_t, u_t, v_t$ to determine all of them. That is a big saving in the number of parameters. At last, our model is interpretable, and combines domain knowledge in finance. For quantitative evaluation, we implement the traditional quantile regression in our experiments, also coupled with an LSTM unit. Mathematically describing it,

in Equation (8), the output $\mu_t, \sigma_t, u_t, v_t$ are replaced by quantiles $q_k^t$: $[q_1^t, \ldots, q_K^t]^\top = \tanh(W^o h_t + b^o)$ and the summed loss $\frac{1}{K} \frac{1}{T-L} \sum_{k=1}^K \sum_{t=L+1}^T L_{\tau_k}(r_t, q_k^t)$ is minimized as in Equation (9). For new subsequent time $t'$, the predicted quantiles $q_1^{t'}, \ldots, q_K^{t'}$ of $r_{t'}$ are sorted to ensure no crossing. We denote this model by LSTM-TQR.

Generally, feature vector sequence $x_1^t, x_2^t, \ldots, x_L^t$ should be designed to contain any information that is related to the conditional distribution of $r_t$ or is helpful to the prediction, like trading volume, related assets, or fundamentals. To keep consistency with GARCH family and ensure the fairness of the comparisons in experiments, we construct $x_1^t, x_2^t, \ldots, x_L^t$ only from past returns $r_{t-1}, r_{t-2}, \ldots$. In real applications of our method, more information can be included in the feature vector sequence.

Our method is widely applicable in quantile prediction or time series modelling in many other non-financial fields. Time series data exhibiting asymmetrical time-varying tail behavior and nonlinear serial dependence of conditional distribution, e.g., hydrologic data, internet traffic data, and electricity price and demand, is most suited. One can also change the standard normal distribution in the Q-Q plot ($Z_\tau$ in HTQF in Equation (6)) to other baseline distribution, to let the HTQF have a controllable shape in the Q-Q plot against the specified distribution, like exponential one or lognormal one, the choice of which relies on domain knowledge.

## 5 Experiments

Our experiments are conducted on three types of time series datasets: simulated data, daily asset returns (of stock indexes, exchange rates, and treasury yields), and intraday 5-minute commodity futures returns. For daily returns, for every time series, the data of maximum possible length is used, e.g., S&P 500 index returns start from January 4, 1950 and end at July 2, 2018, which is the longest series with more than 17,000 observations. The shortest has nearly 8000 observations. For intraday commodity futures returns, the recent 1-year every 5-minute returns are used and each series has about 20,000 observations. All returns are calculated by $r_t = P_t/P_{t-1} - 1$ where $P_t$ is the price, rate, or yield at time $t$.

Each time series is divided into three successive parts, for training, validation, and testing respectively. The training set is four fifths of the original series, and the validation and test sets are both one tenth. The training set is normalized to have sample mean 0 and sample variance 1, followed by normalizing the validation and test sets in the exactly same way. The validation set is used for tuning hyper-parameters, and for stopping training when the loss on the validation set begins to increase, to prevent overfitting. Our model has two hyper-parameters, the length $L$ of past series $r_{t-1}, \ldots, r_{t-L}$ on which time-$t$ HTQF parameters $\mu_t, \sigma_t, u_t, v_t$ depend, and the hidden state dimension $H$ of the LSTM unit. We denote our model with them by LSTM-HTQF($L,H$). Similarly, the LSTM-TQR model described in Section 4.3 also has $L$ and $H$ as hyper-parameters. Our competing models are mainly GARCH-type models, from which we select some popular ones for comparisons: GARCH, GARCH-$t$, EGARCH-$t$, GJR-GARCH-$t$, AR-EGARCH-$t$, and AR-GJR-GARCH-$t$. In all of them, $s, p, q$ are hyper-parameters that will be tuned (see Section 2 for details).

The tuning of the hyper-parameters is done in the following sets: $L \in \{40, 60, 80, 100\}$, $H \in \{8, 16\}$, and $s, p, q \in \{1, 2, 3\}$. The $A$ in the HTQF is set to be 4 arbitrarily. We choose $K = 21$ probability levels into the $\tau$ set: $[\tau_1, \ldots, \tau_{21}] = [0.01, 0.05, 0.1, \ldots, 0.9, 0.95, 0.99]$. Performance is evaluated using the pinball loss on the test set. GARCH-type models can easily do this because the conditional PDF is modelled. For comparisons from different perspectives, two test performances over two different $\tau$ sets are evaluated: one is the full $\tau$ set, the other is $[0.01, 0.05, 0.1]$, the quantiles of which are VaR representing downside risk.

### 5.1 Simulated Data

The purpose of the simulation experiment is to verify whether our method can learn the true temporal behavior of the conditional distribution of a given time series. We generate our simulated time series in a way similar to GARCH-$t$ model, but differently, let the degrees of freedom $\nu_t$ be time-varying. Specifically, starting from $r_0 = 0$ and $\sigma_0 = 1$, the time series $\{r_t\}$ together with the scale parameter $\{\sigma_t\}$ and tail parameter $\{\nu_t\}$ are generated as follows:

$$\nu_t = \max\{8 - 2\pi_t, 3\}, \quad \pi_t = \sqrt{0.136 + 0.257 r_{t-1}^2 + 0.717 \pi_{t-1}^2}, \tag{10}$$

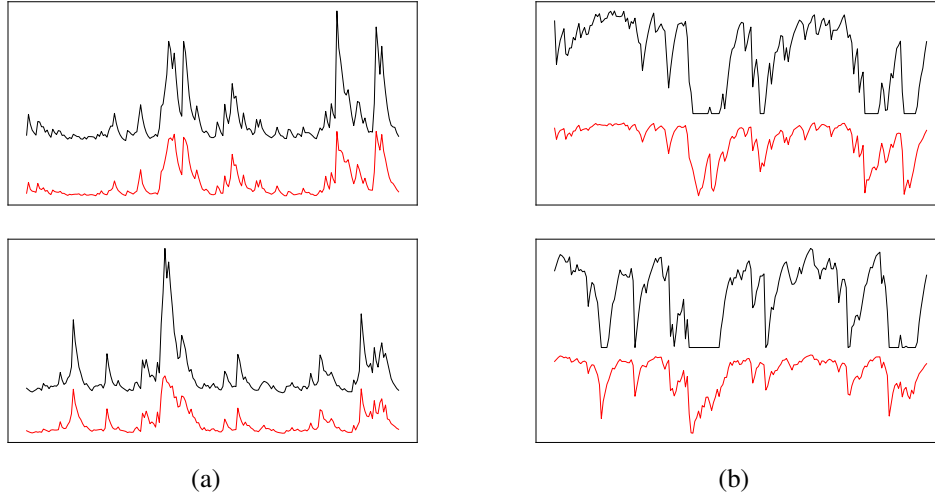

<div align="center">(a)               (b)</div>

Figure 2: Comparisons between true parameters $\{\sigma_t\}$ $\{\nu_t\}$ (black lines) and the learned HTQF parameters $\{\hat{\sigma}_t\}$ $\{\hat{u}_t\}$ (red lines). Upper part of (a): $\{\sigma_t\}$ v.s. $\{\hat{\sigma}_t\}$ on the training set; lower part of (a): $\{\sigma_t\}$ v.s. $\{\hat{\sigma}_t\}$ on the test set; upper part of (b): $\{\nu_t\}$ v.s. $\{\hat{u}_t\}$ on the training set; lower part of (b): $\{\nu_t\}$ v.s. $\{\hat{u}_t\}$ on the test set. Linear transformations are made before plotting.

$$\sigma_t = \sqrt{0.293 + 0.161 r_{t-1}^2 + 0.575 \sigma_{t-1}^2}, \quad r_t = \sigma_t z_t, \quad z_t \text{ is sampled from } t(\nu_t). \tag{11}$$

Totally 10,000 data points are generated. Some example pieces of the generated $\{\sigma_t\}$ and $\{\nu_t\}$ are shown in Figure 2, where the left two black lines are $\{\sigma_t\}$ and the right two black ones are $\{\nu_t\}$. The upper two are from training set, while the lower two are from test set. The red lines are HTQF parameters $\{\hat{\sigma}_t\}$ and $\{\hat{u}_t\}$ learned by our method LSTM-HTQF(20,8) (20 and 8 are set arbitrarily without tuning). $\{\sigma_t\}$ and $\{\hat{\sigma}_t\}$ are plotted together, and $\{\nu_t\}$ and $\{\hat{u}_t\}$ are plotted together. We make linear transformations to the raw quantities to let them be in similar ranges, to be plotted together.

One can see that the learned HTQF scale and tail parameters $\{\hat{\sigma}_t\}$ $\{\hat{u}_t\}$ are highly linearly correlated to the true parameters $\{\sigma_t\}$ $\{\nu_t\}$, on both the training set and the test set. It means that our method has successfully learned the temporal behavior of the conditional distribution of $r_t$. In fact, the linear correlation coefficients between the two lines in the four subplots are 0.8751, -0.8974, 0.9548, and -0.8808 respectively. Negative signs are due to the fact that the heavier the tail, the bigger $\hat{u}_t$ but the smaller $\nu_t$. After running linear regressions between them, we obtain R-squared values: 0.7658, 0.8054, 0.9116, and 0.7758. Another learned parameter $\{\hat{v}_t\}$ is similar to $\{\hat{u}_t\}$, and is not shown here, because the $t$-distribution used for generating the data is symmetric.

## 5.2 Real-world Market Data

In this experiment, first, world's representative stock indexes, exchange rates, and treasury yields are selected, including S&P 500, NASDAQ 100, HSI, Nikkei 225, DAX, FTSE 100, exchange rate of USD to EUR/GBP/CHF/JPY/AUD, and U.S. treasury yield of 2/10/30 years. We report the pinball losses of every methods on the test sets of every asset return series, as shown in Table 1. In parts (a) and (c) of Table 1, the losses over the full $\tau$ set are reported, while in parts (b) and (d) the losses over only $[0.01, 0.05, 0.1]$ (the quantiles of which are VaR) are reported. It is clearly shown that on most assets our LSTM-HTQF outperforms the competitors. Moreover, in parts (b) and (d), the performance improvements are more significant than in (a) and (c), which is consistent with the intuition that the tails are better modelled by our method. Note that the pinball loss is such a measure between observation and quantile that, even for ground truth quantile, the loss is not zero and is bounded by a positive number. So a small decrease in the loss may actually be a substantial improvement. We also conduct the Kupiec's unconditional coverage test [21], the Christoffersen's independence test [6], and the mixed conditional coverage test for backtesting the VaR forecasts of various models, and show the results in the supplementary material (see [8] for a description of details of these statistical tests). To investigate the tail dynamics captured by the LSTM-HTQF model, we plot the HTQF parameters $\{\hat{u}_t\}$ and $\{\hat{v}_t\}$ on the S&P 500 test set in Figure 3 (a), where the blue line is the right tail parameter

Table 1: The pinball losses on the test sets of daily data of stock indexes, exchange rates, and treasury yields. The losses are evaluated over two different $\tau$ sets: (a)(c) [0.01, 0.05, 0.1, ..., 0.9, 0.95, 0.99]; (b)(d) [0.01, 0.05, 0.1]. US$n$Y represents the U.S. treasury yield of $n$ years.

(a)

| Method\Stock Index | S&P 500 | NASDAQ 100 | HSI | Nikkei 225 | DAX | FTSE 100 |
|---|---|---|---|---|---|---|
| GARCH | 0.2316 | 0.1406 | 0.1623 | 0.2868 | 0.1968 | 0.1987 |
| GARCH-$t$ | 0.2314 | 0.1396 | 0.1612 | 0.2855 | 0.1961 | 0.1987 |
| EGARCH-$t$ | 0.2308 | 0.1395 | 0.1611 | 0.2851 | 0.1957 | 0.1983 |
| GJR-GARCH-$t$ | 0.2314 | 0.1396 | 0.1612 | 0.2855 | 0.1961 | 0.1987 |
| AR-EGARCH-$t$ | 0.2304 | 0.1391 | 0.1611 | 0.2847 | 0.1952 | 0.1982 |
| AR-GJR-GARCH-$t$ | 0.2310 | 0.1393 | 0.1612 | 0.2852 | 0.1963 | 0.1981 |
| LSTM-TQR | 0.2325 | **0.1380** | 0.1601 | **0.2822** | 0.1938 | 0.1961 |
| LSTM-HTQF | **0.2299** | 0.1387 | **0.1598** | 0.2854 | **0.1932** | **0.1959** |

(b)

| Method\Stock Index | S&P 500 | NASDAQ 100 | HSI | Nikkei 225 | DAX | FTSE 100 |
|---|---|---|---|---|---|---|
| GARCH | 0.1039 | 0.0669 | 0.0729 | 0.1339 | 0.0853 | 0.0855 |
| GARCH-$t$ | 0.1048 | 0.0667 | 0.0719 | 0.1330 | 0.0850 | 0.0861 |
| EGARCH-$t$ | 0.1037 | 0.0668 | 0.0717 | 0.1324 | 0.0840 | 0.0854 |
| GJR-GARCH-$t$ | 0.1048 | 0.0667 | 0.0719 | 0.1330 | 0.0850 | 0.0861 |
| AR-EGARCH-$t$ | 0.1041 | 0.0666 | 0.0715 | 0.1327 | 0.0834 | 0.0856 |
| AR-GJR-GARCH-$t$ | 0.1052 | 0.0666 | 0.0717 | 0.1333 | 0.0854 | 0.0852 |
| LSTM-TQR | 0.1032 | **0.0644** | 0.0709 | **0.1284** | 0.0812 | 0.0830 |
| LSTM-HTQF | **0.1025** | 0.0646 | **0.0702** | 0.1289 | **0.0810** | **0.0827** |

(c)

| Method\Asset | USDEUR | USDGBP | USDCHF | USDJPY | USDAUD | US2Y | US10Y | US30Y |
|---|---|---|---|---|---|---|---|---|
| GARCH | 0.2260 | 0.2361 | 0.2025 | 0.2222 | 0.2329 | 0.1935 | 0.2861 | 0.2952 |
| GARCH-$t$ | 0.2258 | 0.2366 | 0.2009 | 0.2206 | 0.2338 | 0.1931 | 0.2858 | 0.2948 |
| EGARCH-$t$ | 0.2258 | 0.2352 | 0.2032 | 0.2202 | 0.2370 | 0.1923 | 0.2855 | 0.2940 |
| GJR-GARCH-$t$ | 0.2258 | 0.2366 | 0.2009 | 0.2206 | 0.2338 | 0.1931 | 0.2858 | 0.2948 |
| AR-EGARCH-$t$ | 0.2258 | 0.2353 | 0.2007 | 0.2199 | 0.2367 | **0.1916** | 0.2854 | 0.2941 |
| AR-GJR-$t$ | 0.2259 | 0.2367 | 0.2005 | 0.2203 | 0.2346 | 0.1924 | 0.2857 | 0.2947 |
| LSTM-TQR | 0.2250 | **0.2350** | 0.1966 | 0.2195 | **0.2318** | 0.1928 | 0.2872 | 0.2943 |
| LSTM-HTQF | **0.2247** | 0.2351 | **0.1966** | **0.2193** | 0.2322 | 0.1925 | **0.2849** | **0.2937** |

(d)

| Method\Asset | USDEUR | USDGBP | USDCHF | USDJPY | USDAUD | US2Y | US10Y | US30Y |
|---|---|---|---|---|---|---|---|---|
| GARCH | 0.0942 | 0.0965 | 0.1041 | 0.0996 | 0.0913 | 0.0902 | 0.1232 | 0.1232 |
| GARCH-$t$ | 0.0941 | 0.0984 | 0.1026 | 0.0978 | 0.0923 | 0.0876 | 0.1229 | 0.1236 |
| EGARCH-$t$ | 0.0938 | 0.0959 | 0.1054 | 0.0975 | 0.1062 | 0.0879 | 0.1218 | **0.1223** |
| GJR-GARCH-$t$ | 0.0941 | 0.0984 | 0.1026 | 0.0978 | 0.0923 | 0.0876 | 0.1229 | 0.1236 |
| AR-EGARCH-$t$ | 0.0938 | 0.0960 | 0.1026 | 0.0976 | 0.1053 | 0.0877 | 0.1218 | 0.1224 |
| AR-GJR-$t$ | 0.0941 | 0.0982 | 0.1026 | 0.0980 | 0.0916 | 0.0875 | 0.1229 | 0.1226 |
| LSTM-TQR | **0.0923** | 0.0948 | 0.0975 | 0.0965 | 0.0899 | 0.0879 | 0.1231 | 0.1231 |
| LSTM-HTQF | 0.0930 | **0.0946** | **0.0971** | **0.0958** | **0.0897** | **0.0869** | **0.1199** | 0.1224 |

$\{\hat{u}_t\}$ and the red one is the left $\{\hat{v}_t\}$. We can see roughly similar patterns in the two lines, both with clustering and spikes, but different in details.

At last, we collect intraday 5-minute returns of five commodity futures from Chinese futures market: steel rebar, natural rubber, soybean, cotton, and sugar. To reduce the difficulty, the overnight jumps are eliminated. In the same way as daily asset returns, the losses on the test sets are reported in Table 2, which also shows that our LSTM-HTQF outperforms the competitors on most assets. The plotting of $\{\hat{u}_t\}$ and $\{\hat{v}_t\}$ on the steel rebar test set is shown in Figure 3 (b), which indicates that high-frequency financial asset returns also have time-varying heavy tails. The different tail dynamic with S&P 500 may attribute to the different time scales of the two time series.

## 6   Conclusions

In summary, in this paper, we proposed a parametric HTQF to represent the asymmetric heavy-tailed conditional distribution of financial return series. The dependence of HTQF's parameters on past

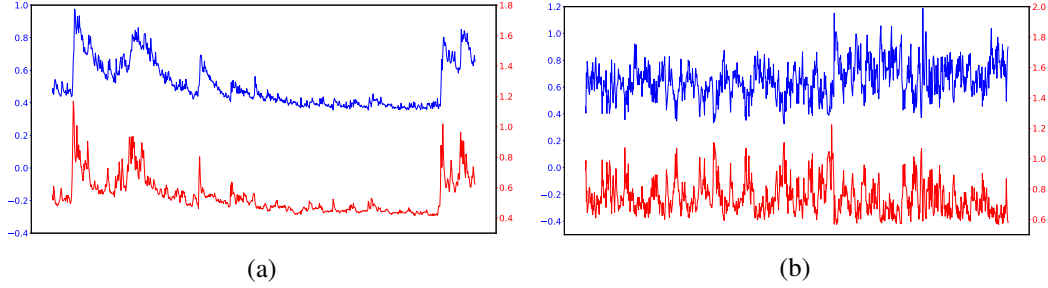

(a)                                                                (b)

Figure 3: The HTQF parameters $\{\hat{u}_t\}$ and $\{\hat{v}_t\}$ on the test set of: (a) S&P 500 daily data; (b) steel rebar 5-minute data. The blue line is $\{\hat{u}_t\}$ and the red one is $\{\hat{v}_t\}$.

Table 2: The pinball losses on the test sets of 5-minute return data of commodity futures. The losses are evaluated over two different $\tau$ sets: (a) [0.01, 0.05, 0.1, ..., 0.9, 0.95, 0.99]; (b) [0.01, 0.05, 0.1].

(a)

| Method\Commodity | Steel Rebar | Natural Rubber | Soybean | Cotton | Sugar |
|---|---|---|---|---|---|
| GARCH | 0.1770 | 0.1701 | 0.2424 | 0.1621 | 0.1958 |
| EGARCH-t | 0.1643 | 0.1564 | 0.2392 | 0.1524 | 0.1859 |
| GJR-GARCH-t | 0.1648 | 0.1576 | 0.2393 | 0.1526 | 0.1859 |
| AR-EGARCH-t | 0.1646 | 0.1572 | 0.2391 | 0.1522 | 0.1857 |
| AR-GJR-GARCH-t | 0.1652 | 0.1586 | 0.2391 | 0.1524 | 0.1857 |
| LSTM-TQR | 0.1644 | **0.1543** | 0.2389 | 0.1504 | 0.1844 |
| LSTM-HTQF | **0.1639** | 0.1548 | **0.2385** | **0.1501** | **0.1842** |

(b)

| Method\Commodity | Steel Rebar | Natural Rubber | Soybean | Cotton | Sugar |
|---|---|---|---|---|---|
| GARCH | 0.0882 | 0.0885 | 0.1077 | 0.0783 | 0.0994 |
| EGARCH-t | 0.0797 | 0.0797 | 0.1062 | 0.0720 | 0.0935 |
| GJR-GARCH-t | 0.0801 | 0.0807 | **0.1059** | 0.0721 | 0.0935 |
| AR-EGARCH-t | 0.0805 | 0.0810 | 0.1063 | 0.0719 | 0.0937 |
| AR-GJR-GARCH-t | 0.0807 | 0.0825 | 0.1060 | 0.0721 | 0.0937 |
| LSTM-TQR | 0.0769 | **0.0765** | **0.1059** | 0.0710 | 0.0922 |
| LSTM-HTQF | **0.0767** | 0.0770 | 0.1065 | **0.0704** | **0.0916** |

information is modelled by an LSTM unit. The pinball loss between the observation and conditional quantiles makes the learning of LSTM parameters be in a quantile regression framework, which overcomes the disadvantages of traditional quantile regression. After learning, conditional quantiles or VaR can be predicted with relatively better accuracy, and besides, the plotting of HTQF parameters shows us the dynamic tail behaviors of financial asset returns, some of which display clustering and spikes but difference between left and right tails.

Although our paper focuses on the tail dynamics, in the future, more advanced models that can learn more elaborate dynamics of the conditional distribution of financial time series are necessary, e.g., improving the flexibility of the HTQF or modifying the way how LSTM is used may be needed. Moreover, it is important to discover how LSTM and HTQF in our model work respectively and how they contribute to the performance improvements. It is also interesting to interpret what tail dynamics of financial assets we have captured, and what consequences it has for understanding market behaviors, for asset pricing, and for risk management. To make those clear, we may need more in-depth analysis of our model and more statistical testing and analysis of the VaR or quantile forecasts of the model in the future.

## Acknowledgement

Qi WU acknowledges the financial support from the Hong Kong Research Grants Council, in particular the Early Career Scheme 24200514 and the General Research Funds 14211316 and 14206117.

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
