[Supplementary Material · NeurIPS2018_XingYan_supplementary.pdf]

# Parsimonious Quantile Regression of Financial Asset Tail Dynamics via Sequential Learning

**Xing Yan[3]**     **Weizhong Zhang[1]**     **Lin Ma[1]**     **Wei Liu[1]**     **Qi Wu[2,*]**
[1]Tencent AI Lab
[2]School of Data Science, City University of Hong Kong
[3]Department of SEEM, The Chinese University of Hong Kong
xyan@se.cuhk.edu.hk  {zhangweizhongzju,forest.linma}@gmail.com
wl2223@columbia.edu  qiwu55@cityu.edu.hk

## Supplementary Material

## Appendix A. The Proof of the Existence of HTQF's Unique Probability Distribution

The proof idea is to show that HTQF is continuously differentiable, is strictly monotonically increasing over $(0, 1)$, and approaches $-\infty/+\infty$ as $\tau$ tends to $0/1$. So the inverse function of HTQF exists and is a cumulative distribution function.

The HTQF has the specification:

$$Q(\tau|\mu, \sigma, u, v) = \mu + \sigma Z_\tau(\frac{e^{uZ_\tau}}{A} + 1)(\frac{e^{-vZ_\tau}}{A} + 1) = \mu + \sigma g(Z_\tau), \tag{1}$$

where $g(x) = x(\frac{e^{ux}}{A} + 1)(\frac{e^{-vx}}{A} + 1)$. $Z_\tau$ is the quantile function of the standard normal distribution, so we only need to prove that $g(x)$ is continuously differentiable, is strictly monotonically increasing over $(-\infty, +\infty)$, and approaches $-\infty/+\infty$ as $x$ tends to $-\infty/+\infty$. Obviously $g(x)$ is continuously differentiable and $\lim_{x\to-\infty/+\infty} g(x) = -\infty/+\infty$. To prove the monotonicity, we calculate the derivative of $g(x)$:

$$g'(x) = (\frac{e^{ux}}{A} + 1)(\frac{e^{-vx}}{A} + 1) + xu\frac{e^{ux}}{A}(\frac{e^{-vx}}{A} + 1) - xv(\frac{e^{ux}}{A} + 1)\frac{e^{-vx}}{A} \tag{2}$$

$$= \frac{e^{(u-v)x}}{A^2}(1 + (u-v)x) + \frac{e^{ux}}{A}(1 + ux) + \frac{e^{-vx}}{A}(1 - vx) + 1 \tag{3}$$

$$= \frac{1}{A^2}h((u-v)x) + \frac{1}{A}h(ux) + \frac{1}{A}h(-vx) + 1. \qquad (h(x) = e^x(1 + x)) \tag{4}$$

Next we prove $h(x) \geq -1, \forall x$. This is equivalent to $1 + x \geq -e^{-x}$, or $1 + x + e^{-x} \geq 0, \forall x$. A simple monotonic analysis on the function $1 + x + e^{-x}$ can reveal that its global minimum is reached at $x = 0$, so $1 + x + e^{-x} \geq 2 \geq 0$. So, $h(x) \geq -1$ and

$$g'(x) \geq -\frac{1}{A^2} - \frac{1}{A} - \frac{1}{A} + 1. \tag{5}$$

If we choose $A \geq 3$, then $g'(x) \geq -\frac{1}{9} - \frac{1}{3} - \frac{1}{3} + 1 = \frac{2}{9} > 0$ holds for all $x$. So $g(x)$ is strictly monotonically increasing and our proof is completed.

---

## Appendix B. A Brief Introduction to LSTM

In machine learning, LSTM is a type of recurrent neural network model designed for capturing both long-term and short-term dependencies or complex dynamics in sequential data. Recently remarkable successes have been made in its applications like speech recognition, machine translation, protein structure prediction, etc. Mathematically, LSTM is a highly composite nonlinear parametric function that maps a sequence of vectors $x_1, \ldots, x_n$ to another sequence of vectors $y_1, \ldots, y_n$ (or to just one vector $y$), through hidden state vectors $h_1, \ldots, h_n$. Examples include machine translation from a Chinese sentence to an English sentence, and the classification of a music clip to its genre.

Before describing the full mathematics of it, we first introduce the simple recurrent neural network model, which is the understructure of LSTM and has the functional form:

$$h_j = \sigma_h(W_h x_j + U_h h_{j-1} + b_h), \tag{6}$$

$$y_j = \sigma_y(W_y h_j + b_y). \tag{7}$$

$W_h, U_h, b_h, W_y, b_y$ are the parameters and $\sigma_h, \sigma_y$ are nonlinear activation functions. It models a nonlinear functional relationship between $x_1, \ldots, x_n$ and $y_1, \ldots, y_n$. One can stack this structure multiple times to get multi-layered or hierarchical recurrent neural network, which is a type of deep learning model.

LSTM extends this structure and has the equations:

$$f_j = \sigma_g(W_f x_j + U_f h_{j-1} + b_f), \tag{8}$$

$$i_j = \sigma_g(W_i x_j + U_i h_{j-1} + b_i), \tag{9}$$

$$o_j = \sigma_g(W_o x_j + U_o h_{j-1} + b_o), \tag{10}$$

$$g_j = \sigma_h(W_g x_j + U_g h_{j-1} + b_g), \tag{11}$$

$$c_j = f_j * c_{j-1} + i_j * g_j, \tag{12}$$

$$h_j = o_j * \sigma_h(c_j), \tag{13}$$

and at last, the output $y_j$ is a nonlinear function of $h_j$, like:

$$y_j = \sigma_h(W_y h_j + b_y). \tag{14}$$

All $W, U, b$ are parameters and $\sigma_g, \sigma_h$ are nonlinear activation functions, which are chosen as logistic function and tanh function respectively in this paper. $*$ represents element-wise multiplication of two vectors. In the case of outputting only one vector $y$, one can use the average of all hidden state vectors $\frac{1}{n} \sum h_j$ or just the last one $h_n$, e.g., $y = \sigma_h(W_y h_n + b_y)$, like the Equation (8) in our paper.

The tanh function and logistic function are the two most popular activation functions in neural network models, playing a role of nonlinear transformation with finite range. Multiple compositions of these activation functions can approximate complex nonlinear relationships between input vectors $x_1, \ldots, x_n$ and output vectors $y_1, \ldots, y_n$ or $y$.

## Appendix C. Statistical Testing of VaR Forecasts

Consider a sequence of realized returns or observations $\{r_{t'}\}$ and a sequence of VaR or quantile forecasts $\{q^{t'}\}$ of a fixed probability level $\tau$ by any model. In order to implement the testing procedure, we need the definition of hitting sequence of quantile violations:

$$I_{t'} = \begin{cases} 1 & \text{if } r_{t'} < q^{t'} \\ 0 & \text{if } r_{t'} \geq q^{t'} \end{cases} . \tag{15}$$

Ideally, $\{I_{t'}\}$ should be an i.i.d. Bernoulli distribution sequence with parameter $\tau$. To test that, Kupiec's unconditional coverage test [3] checks if the unconditional distribution of $\{I_{t'}\}$ is the Bernoulli distribution, i.e., if the proportion of quantile violations is equal to $\tau$. Christoffersen's independence test [1] checks if $I_{t'}$ is independent of $I_{t'-1}$, i.e., current violation (or not) is independent of previous violation (or not). The mixed conditional coverage test jointly checks these two null hypotheses. One can refer to [2] for details of the three tests. We report the statistics of these tests for $\tau = 0.01, 0.05, 0.1$ VaR forecasts in the following tables. In some sense, the smaller the test statistic, the better the forecasts. The results show that our LSTM-HTQF model does quite well in some cases.

Table 1: Unconditional coverage test for $\tau = 0.01$ quantile forecasts. The test statistic shown below has a chi-square distribution with one degree of freedom. The threshold for rejecting the null hypothesis with 95% confidence level is 3.8415. $^*$ represents the threshold is exceeded. In the parentheses, we report the number of quantile violations given by each model against the ideal number of violations.

(a)

| Method\Stock Index | S&P 500 | NASDAQ 100 | HSI | Nikkei 225 | DAX | FTSE 100 |
|---|---|---|---|---|---|---|
| GARCH | 14.4440* (35/17) | 4.6253* (15/8) | 2.0955 (12/8) | 21.5459* (33/13) | 2.1763 (12/8) | 6.2953* (17/9) |
| GARCH-$t$ | 3.9938* (26/17) | 4.6253* (15/8) | 1.0776 (5/8) | 3.1830 (20/13) | 0.6900 (10/8) | 1.8839 (13/9) |
| EGARCH-$t$ | 5.8339* (28/17) | 8.8965* (18/8) | 0.0627 (7/8) | 6.1990* (23/13) | **0.2423** (9/8) | 3.8175 (15/9) |
| GJR-GARCH-$t$ | 3.9938* (26/17) | 4.6253* (15/8) | 1.0776 (5/8) | 3.1830 (20/13) | 0.6900 (10/8) | 1.8839 (13/9) |
| AR-EGARCH-$t$ | 6.8642* (29/17) | 8.8965* (18/8) | 0.0627 (7/8) | 7.3869* (24/13) | **0.2423** (9/8) | 6.2953* (17/9) |
| AR-GJR-GARCH-$t$ | 6.8642* (29/17) | 5.9239* (16/8) | 1.0776 (5/8) | 6.1990* (23/13) | 0.6900 (10/8) | 2.7780 (14/9) |
| LSTM-TQR | 9.1330* (31/17) | **0.0036** (8/8) | **0.0133** (8/8) | 11.4468* (27/13) | 2.0919 (4/8) | **0.0553** (8/9) |
| LSTM-HTQF | 4.8762* (27/17) | 0.1779 (7/8) | 2.1592 (4/8) | **1.6731** (18/13) | 0.3710 (6/8) | 3.1876 (4/9) |

(b)

| Method\Asset | USDEUR | USDGBP | USDCHF | USDJPY | USDAUD | US2Y | US10Y | US30Y |
|---|---|---|---|---|---|---|---|---|
| GARCH | 1.9494 (16/11) | **0.0395** (13/12) | **0.1439** (11/12) | 5.1291* (21/12) | 0.3603 (10/12) | 4.6935* (15/8) | 2.6422 (13/8) | 1.7514 (12/8) |
| GARCH-$t$ | 0.1080 (10/11) | 1.6210 (17/12) | 1.7326 (8/12) | **1.0269** (16/12) | 0.6968 (15/12) | 0.1665 (7/8) | 0.4629 (10/8) | **0.1214** (9/8) |
| EGARCH-$t$ | 0.1080 (10/11) | 0.9862 (9/12) | 4.0185* (6/12) | 2.7314 (7/12) | Inf (0/12) | 1.4109 (5/8) | 0.4629 (10/8) | 1.0174 (11/8) |
| GJR-GARCH-$t$ | 0.1080 (10/11) | 1.6210 (17/12) | 1.7326 (8/12) | **1.0269** (16/12) | 0.6968 (15/12) | 0.1665 (7/8) | 0.4629 (10/8) | **0.1214** (9/8) |
| AR-EGARCH-$t$ | 0.1080 (10/11) | 0.9862 (9/12) | 5.6421* (5/12) | 2.7314 (7/12) | Inf (0/12) | 1.4109 (5/8) | **0.1188** (9/8) | 0.4679 (10/8) |
| AR-GJR-$t$ | 0.1080 (10/11) | 1.6210 (17/12) | 0.4641 (10/12) | **1.0269** (16/12) | 3.4917 (19/12) | 0.1665 (7/8) | 0.4629 (10/8) | 0.1318 (7/8) |
| LSTM-TQR | 0.0768 (12/11) | 0.1439 (11/12) | 5.6421* (5/12) | 4.0185* (6/12) | **0.0803** (13/12) | 0.4048 (10/8) | 1.7413 (12/8) | 2.6548 (13/8) |
| LSTM-HTQF | **0.0004** (11/11) | 0.9862 (9/12) | 2.7314 (7/12) | 2.7314 (7/12) | 0.0883 (11/12) | **0.0021** (8/8) | 1.0098 (11/8) | 3.7148 (14/8) |

Table 2: Independence test for $\tau = 0.01$ quantile forecasts. The test statistic shown below has a chi-square distribution with one degree of freedom. The threshold for rejecting the null hypothesis with 95% confidence level is 3.8415. $^*$ represents the threshold is exceeded.

(a)

| Method\Stock Index | S&P 500 | NASDAQ 100 | HSI | Nikkei 225 | DAX | FTSE 100 |
|---|---|---|---|---|---|---|
| GARCH | 4.3485$^*$ | 4.9017$^*$ | 0.3815 | 1.3335 | 1.8207 | 0.9185 |
| GARCH-$t$ | 12.3925$^*$ | 4.9017$^*$ | 0.0656 | 1.1056 | 0.2667 | 1.7643 |
| EGARCH-$t$ | 6.6850$^*$ | 7.7473$^*$ | 0.1289 | 0.7068 | 0.2157 | 1.2932 |
| GJR-GARCH-$t$ | 12.3925$^*$ | 4.9017$^*$ | 0.0656 | 1.1056 | 0.2667 | 1.7643 |
| AR-EGARCH-$t$ | 6.3010$^*$ | 7.7473$^*$ | 0.1289 | 0.5980 | 0.2157 | 0.9185 |
| AR-GJR-GARCH-$t$ | 10.6726$^*$ | 4.4169$^*$ | 0.0656 | 0.7068 | 0.2667 | 1.5153 |
| LSTM-TQR | 5.5872$^*$ | **3.4722** | 0.1686 | 1.1391 | **0.0423** | 3.5864 |
| LSTM-HTQF | **3.2329** | 4.0133$^*$ | **0.0419** | **0.5027** | 4.5182$^*$ | **0.0371** |

(b)

| Method\Asset | USDEUR | USDGBP | USDCHF | USDJPY | USDAUD | US2Y | US10Y | US30Y |
|---|---|---|---|---|---|---|---|---|
| GARCH | 0.4697 | 2.3621 | 0.1987 | 0.8081 | **0.1681** | 0.5647 | 0.4295 | 0.3660 |
| GARCH-$t$ | **0.1825** | 1.4367 | 0.1048 | 1.6354 | 0.3798 | 0.1217 | 0.2532 | 0.2051 |
| EGARCH-$t$ | **0.1825** | 3.7741 | 0.0589 | 0.0802 | Inf | **0.0620** | 0.2532 | 0.3071 |
| GJR-GARCH-$t$ | **0.1825** | 1.4367 | 0.1048 | 1.6354 | 0.3798 | 0.1217 | 0.2532 | 0.2051 |
| AR-EGARCH-$t$ | **0.1825** | 3.7741 | **0.0408** | 0.0802 | Inf | **0.0620** | **0.2048** | 0.2535 |
| AR-GJR-$t$ | **0.1825** | 1.4367 | 0.1641 | 1.6354 | 0.6114 | 0.1217 | 0.2532 | **0.1237** |
| LSTM-TQR | 0.2633 | 0.1987 | **0.0408** | **0.0589** | 0.2848 | 0.2494 | 0.3655 | 0.4300 |
| LSTM-HTQF | 0.2210 | **0.1328** | 0.0802 | 0.0802 | 0.2035 | 0.1592 | 0.3067 | 0.4994 |

Table 3: Conditional coverage test for $\tau = 0.01$ quantile forecasts. The test statistic shown below has a chi-square distribution with two degree of freedom. The threshold for rejecting the null hypothesis with 95% confidence level is 5.9915. $^*$ represents the threshold is exceeded.

(a)

| Method\Stock Index | S&P 500 | NASDAQ 100 | HSI | Nikkei 225 | DAX | FTSE 100 |
|---|---|---|---|---|---|---|
| GARCH | 18.7924$^*$ | 9.5270$^*$ | 2.4770 | 22.8794$^*$ | 3.9970 | 7.2138$^*$ |
| GARCH-$t$ | 16.3863$^*$ | 9.5270$^*$ | 1.1432 | 4.2886 | 0.9567 | 3.6482 |
| EGARCH-$t$ | 12.5188$^*$ | 16.6438$^*$ | 0.1916 | 6.9058$^*$ | **0.4580** | 5.1107 |
| GJR-GARCH-$t$ | 16.3863$^*$ | 9.5270$^*$ | 1.1432 | 4.2886 | 0.9567 | 3.6482 |
| AR-EGARCH-$t$ | 13.1652$^*$ | 16.6438$^*$ | 0.1916 | 7.9849$^*$ | **0.4580** | 7.2138$^*$ |
| AR-GJR-GARCH-$t$ | 17.5368$^*$ | 10.3408$^*$ | 1.1432 | 6.9058$^*$ | 0.9567 | 4.2934 |
| LSTM-TQR | 14.7202$^*$ | **3.4758** | **0.1819** | 12.5860$^*$ | 2.1342 | 3.6416 |
| LSTM-HTQF | 8.1091$^*$ | 4.1912 | 2.2011 | **2.1758** | 4.8892 | **3.2247** |

(b)

| Method\Asset | USDEUR | USDGBP | USDCHF | USDJPY | USDAUD | US2Y | US10Y | US30Y |
|---|---|---|---|---|---|---|---|---|
| GARCH | 2.4192 | 2.4016 | **0.3426** | 5.9372 | 0.5284 | 5.2581 | 3.0717 | 2.1174 |
| GARCH-$t$ | 0.2905 | 3.0578 | 1.8375 | **2.6623** | 1.0766 | 0.2882 | 0.7161 | 0.3264 |
| EGARCH-$t$ | 0.2905 | 4.7603 | 4.0773 | 2.8116 | Inf | 1.4729 | 0.7161 | 1.3245 |
| GJR-GARCH-$t$ | 0.2905 | 3.0578 | 1.8375 | **2.6623** | 1.0766 | 0.2882 | 0.7161 | 0.3264 |
| AR-EGARCH-$t$ | 0.2905 | 4.7603 | 5.6829 | 2.8116 | Inf | 1.4729 | **0.3237** | 0.7214 |
| AR-GJR-$t$ | 0.2905 | 3.0578 | 0.6281 | **2.6623** | 4.1031 | 0.2882 | 0.7161 | **0.2556** |
| LSTM-TQR | 0.3401 | **0.3426** | 5.6829 | 4.0773 | 0.3650 | 0.6542 | 2.1068 | 3.0849 |
| LSTM-HTQF | **0.2215** | 1.1190 | 2.8116 | 2.8116 | **0.2918** | **0.1613** | 1.3165 | 4.2142 |

Table 4: Unconditional coverage test for $\tau = 0.05$ quantile forecasts. The test statistic shown below has a chi-square distribution with one degree of freedom. The threshold for rejecting the null hypothesis with 95% confidence level is 3.8415. $^*$ represents the threshold is exceeded. In the parentheses, we report the number of quantile violations given by each model against the ideal number of violations.

(a)