[Reviews · NeurIPS 2018]

Reviewer 1



Summary This paper describes an approach to learning the dynamics of financial time series. The authors describe a parametric quantile function with four parameters (modelling location, scale, and the shapes of the left and right hand tails of the conditional distribution of returns). The time dynamics of these parameters are learned using LSTM neural network. The performance of the algorithm is compared to various GARCH-type specifications and a TQR model (which combines "traditional" quantile regression with a LTSM neural network). Strengths I enjoyed reading the paper. The ideas are nicely conveyed (although there are some problems with the grammar). The idea of using a parametric quantile function offers a way to move beyond models which concentrate on volatility and allows time-varying estimation of tails. The idea of combining this approach with a LTSM is a novel and interesting specification which allows complex dynamics in the conditional distribution of returns. The performance is compared to a range of generally used models. The authors make a good case that their model leads to better estimation of the time-variation in the tails of the conditional distribution of the returns. Weaknesses My main concern is whether the Q(tau|.) function in equation (6) is sufficiency flexible since the tails have to vary in a particular parametric way (which depends on the values of u and v). I would have preferred a model where the complexity of (6) could be tuned in the learning algorithm. I don't understand the claim that the model is easier to fit than GARCH-type models since both involve optimisation of some loss function. The authors show improved performance over GARCH-type models. I found it difficult to understand whether these improvements are "substantial". The paper also lacks discussion about where the improvements come from: is it from the LTSM or quantile function? TQR and QR-HTQF often have very similar performance and so I would suggest that it's the LTSM which is helping (in particular, TQR may prefer better if the number of quantile used was optimised in some way). Confidence score I have a lot of experience working on modelling financial time series in econometrics.

Reviewer 2



The paper proposes to express the distribution of a real-valued univariate random variable as a tail deviation from the quantile function of a normal variate, called HTQF. This function consists of 4 parameters to specify the location, scale, lower excess tail, and upper excess tail. In a time series context, the paper models these quantities at time t in a time-varying fashion through an LSTM, using a window of L past observations (and higher central moments). Since the CDF of the HTQF model is monotonically increasing, the quantile crossing problem is avoided by construction. The model is trained by minimizing the set of pinball losses at the various quantiles of interest. The paper is quite well written and easy to follow. The proposed form for the HTQF is interesting and appears to be a novel contribution worthy of a presentation at NIPS. The experimental validation on synthetic and real datasets is adequate, and the latter appears to show some improvement over well-known GARCH models. In addition to the pinball loss, it would have been helpful to provide coverage measures for the quantiles (i.e. in a properly-calibrated 0.1 quantile, 10% of the observations should fall below it; what’s the empirical quantile corresponding to the nominal model quantile? This would be akin to a Q-Q plot between the the realized quantiles and the predicted quantiles). This measure would also more adequately compare the behavior of various models across all quantiles, which are not systematically reported in the results. Moreover, a small quibble would be that no model from the stochastic volatility family are represented, which are the other large family of volatility models in financial econometrics apart from GARCH. The main issue that I have with the approach is that the specification of the excess tails takes a very constrained functional form, which limits its flexibility — it would be interesting to understand whether this leads to underfitting for the financial asset return series of interest. It is also obvious that this approach can significantly enhance time series modeling in many other areas beyond finance; I would be looking forward to a discussion of which fields would appear most suited to benefit from this approach. Overall, I think that the issues reported above are fixable, and if so I would be happy to see the paper accepted. Detailed comments: * Line 18: care ==> care about * Line 27: gaussian ==> Gaussian * Line 41: speak itself ==> speak for itself * Line 137: has ==> have * Line 236: Verity ==> Verify * Line 236: truth ==> true * Line 249: truth ==> true * Line 249: no matter it is on ==> on both * Paragraph 248-255: this analysis is sketchy and should be made more precise.

Reviewer 3



This paper proposes a new regression framework to learn time-varying quantiles of asymmetric heavy-tailed financial asset returns. The methodology is based on the novel heavy-tailed quantile function (HTQF) that models the conditional quantile with capturing how the tail-behavior is far from (or close to) a Gaussian one. The (time-varying) parameters of this function are generated as outputs of long short-term memory (LSTM), where the inputs are polynomials of the centered return processes. Following the quantile regression procedure, the model is finally estimated. The authors provide thorough experiments and real data analysis in the latter half of the paper. The methodology is unique and interesting in that the traditional quantile regression framework is combined with a recent technique of LSTM and the original HTQF. Comments: - It's better to cite CAViaR paper by Engle and Manganelli (2004, Journal of Business & Economic Statistics), which proposed a bit similar methodology to estimate quantiles inspired by GARCH models. - The proposed framework will become clearer if eq. (8) is explained more specifically. Readers from the economics side are not necessarily familiar with LSTM. (More precisely, I’m curious why (8) can generate the parameters successfully. Why is tanh used in the last step? I’m happy to get intuitive explanation as I’m from the economics side.) - On page4, the authors write it can be shown that, for an HTQF with fixed values of its parameters, there exists a unique probability distribution associated with it.’’ A formal proof will help readers to understand it.